# Diagnostic Performance of Conventional X-ray for Detecting Foreign Bodies in the Upper Digestive Tract: A Systematic Review and Diagnostic Meta-Analysis

**DOI:** 10.3390/diagnostics11050790

**Published:** 2021-04-27

**Authors:** Ta-Wei Yang, Yi-Chung Yu, Yen-Yue Lin, Shih-Chang Hsu, Karen Chia-Wen Chu, Chin-Wang Hsu, Chyi-Huey Bai, Cheng-Kuang Chang, Yuan-Pin Hsu

**Affiliations:** 1Department of Emergency Medicine, Taoyuan Armed Forces General Hospital, Taoyuan 325, Taiwan; taweiyang@gmail.com (T.-W.Y.); yyline.tw@yahoo.com.tw (Y.-Y.L.); 2National Defense Medical Center, Taipei 114, Taiwan; 3Emergency Department, Wan Fang Hospital, Taipei Medical University, Taipei 116, Taiwan; xx0011tw@hotmail.com (Y.-C.Y.); 1980bradhsu@gmail.com (S.-C.H.); karencwchu@gmail.com (K.C.-W.C.); wan11119@tmu.edu.tw (C.-W.H.); 4Department of Emergency, School of Medicine, College of Medicine, Taipei Medical University, Taipei 110, Taiwan; 5Department of Public Health, School of Medicine, College of Medicine, Taipei Medical University, Taipei 110, Taiwan; baich@tmu.edu.tw; 6Department of Radiology, Tri-Service General Hospital, National Defense Medical Center, Taipei 114, Taiwan; changlight3@gmail.com; 7Research Center of Big Data and Meta-Analysis, Wan Fang Hospital, Taipei Medical University, Taipei 116, Taiwan

**Keywords:** foreign bodies, fish bone, upper digestive tract, plain radiography

## Abstract

Foreign body (FB) ingestion is a common clinical problem in acute settings. Detecting FBs in the upper digestive tract is challenging. The conventional X-ray is usually the first-line imaging tool to detect FBs. However, its diagnostic performance is inconsistent in the literature. In this study, we performed a systematic review and meta-analysis to determine the diagnostic performance of the conventional X-ray for detecting FBs in the upper digestive tract. We conducted a systematic search of PubMed, Embase, Cochrane Library, Web of Science, and Scopus until 1 August 2020. Prospective or retrospective studies investigating the diagnostic accuracy of conventional X-rays for detecting FBs in the upper digestive tract in patients of all ages were included. The Quality Assessment of Studies of Diagnostic Accuracy-2 tool was used to review the quality of included studies. We used a bivariate random-effects model to calculate diagnostic accuracy parameters. Heterogeneity was assessed using I^2^ statistics. We included 17 studies (*n* = 4809) in the meta-analysis. Of the 17 studies, most studies were rated as having a high risk of bias. Conventional X-rays had a pooled sensitivity of 0.58 (95% confidence interval [CI] = 0.36–0.77, I^2^ = 98.52) and a pooled specificity of 0.94 (95% CI = 0.87–0.98, I^2^ = 94.49) for detecting FBs in the upper digestive tract. The heterogeneity was considerable. The area under the summary receiver operating characteristic curve was 0.91 (95% CI = 0.88–0.93). Deek’s funnel plot asymmetry test results revealed no significant publication bias (*p* = 0.41). The overall sensitivity and specificity of conventional X-rays were low and high, respectively, for detecting FBs in the upper digestive tract. Hence, conventional X-rays to exclude patients without upper FBs in the digestive tract are not recommended. Further imaging or endoscopic examinations should be performed for at-risk patients.

## 1. Introduction

Foreign body (FB) ingestion is a common clinical problem in acute settings. More than 93,000 cases of FB ingestion were reported in the United States in 2018 [1]. Most cases of FB ingestion occur in the pediatric population, with the highest incidence observed between the age of 6 months and 6 years [2]. In adults, FB ingestion is more frequently observed in elder persons with impaired swallowing controls, individuals with underlying psychiatric diseases, or those with alcohol intoxication [3]. Most ingested FBs obstruct the oropharynx and hypopharynx; obstruction of the esophagus by FBs is less common [4]. In regards to the management of ingested FBs, approximately 80–90% of ingested FBs pass through the gastrointestinal system and are excreted without any intervention being required, 10–20% require endoscopic removal, and less than 1% require surgical removal [5]. In addition, a timely and accurate detection of FBs in the upper digestive tract is crucial because undetected FBs present in the esophagus might increase morbidity and even mortality [6]. Therefore, FBs should be removed within 24 h of ingestion because the risk of complications substantially increases over time [5].

Detecting FBs present in the upper digestive tract is challenging for clinicians. The conventional X-ray is usually used as the first-line imaging modality to detect FBs. Although several studies [4,7,8,9,10,11,12,13,14,15,16,17,18,19,20,21,22] have investigated the use of conventional X-rays for detecting FBs, they have reported inconsistent results in terms of diagnostic performance, with sensitivity ranging from 15% to more than 90% [7,19] and specificity ranging from 50% to 100% [7,8]. The American Society for Gastrointestinal Endoscopy and the European Society for Gastrointestinal Endoscopy (ESGE) guidelines strongly recommend using conventional X-rays to detect the presence, location, size, configuration, and the number of ingested FBs if the ingestion of radiopaque objects is suspected or if the type of object is unknown. In addition, the ESGE guideline does not recommend radiological evaluation for patients with nonbony food bolus impaction without complications [2,5]. However, the quality of evidence for these recommendations is low.

Because inconsistent results have been reported in the literature and these findings have not been previously synthesized through a meta-analysis, we performed a systematic review and meta-analysis to evaluate the diagnostic performance of conventional X-rays for detecting FBs in the upper digestive tract.

## 2. Materials and Methods

The protocol of this systematic review and meta-analysis is registered on PROSPERO (PROSPERO ID: CRD42020201034). This study adhered to the Preferred Reporting Items for Systematic Reviews and Meta-Analyses statement [23]. We systematically searched the following databases from their inception until 1 August 2020: PubMed, Embase, Cochrane Library, Web of Science, and Scopus. We used the following keywords to search for relevant studies: X-ray, plain radiography, plain film, FB ingestion, and FB obstruction. The details of our search strategy are listed in Appendix A.

We included both prospective or retrospective studies examining the diagnostic accuracy of conventional X-rays for detecting FBs in the upper digestive tract in patients of all ages. We excluded reviews, case series, case reports, conference proceedings, and animal studies. No language restriction was imposed. Two reviewers (T.W.Y. and Y.C.Y.) independently screened all titles and abstracts to identify potentially eligible studies. The full text of potentially eligible articles was retrieved and checked for inclusion by the two reviewers. If no consensus was reached between the two reviewers, a third reviewer (Y.P.H.) made the final decision. We conducted a study selection using EndNote version 17 (Thomson Research Soft, Stamford, CT, USA). Finally, we checked the reference lists of all included studies to identify additional relevant studies.

Two investigators (K.C.W.C. and S.C.H.) independently extracted data from the included studies. The following data were extracted from each selected study: the name of the first author; publication year; study design; country; inclusion and exclusion criteria; sample size; participants’ age and sex; characteristics of the index test; reference standard; the number of true positive, false positive, false negative, and true negative cases.

Two researchers (T.W.Y. and Y.C.Y.) used the Quality Assessment of Studies of Diagnostic Accuracy-2 (QUADAS-2) to independently assess the quality of included studies [24]. This tool has four domains: patient selection, index test, reference standard, and flow and timing. The risk of bias and concerns regarding the applicability, except for timing domains, and the flow were assessed and rated as low, high, and unclear risk. We summarized the results using the Review Manager version 5.3 (The Nordic Cochrane Center, The Cochrane Collaboration, Copenhagen, Denmark). Disagreements were resolved through a discussion.

We adopted a bivariate random-effects model to calculate the following variables for examining the accuracy of the diagnostic test: sensitivity, specificity, positive likelihood ratio (PLR), negative likelihood ratio (NLR), and diagnostic odds ratio (DOR). In addition, we calculated the area under the summary receiver operating characteristic (SROC) curve. All data were calculated using 95% confidence intervals (CIs). We assessed heterogeneity by using the chi-square test and I^2^ statistics. A *p*-value of <0.1 or I^2^ value of >50% suggested substantial heterogeneity. If substantial heterogeneity was identified, we performed a subgroup analysis based on the following parameters: continents (Asia or outside Asia), including only patients with suspected fish bone ingestion (yes or no), including only patients with suspected esophageal FBs (yes or no), setting (only emergency department [ED] or not only ED), study design (prospective or non-prospective), and age (adult or pediatric). Furthermore, we performed a sensitivity analysis according to sample sizes (≥100 or <100). The publication bias in the meta-analysis of studies examining the accuracy of conventional X-ray was assessed using Deeks’ funnel plot asymmetry test [25]. We performed the meta-analysis by using the MIDAS module for StataMP version 14 (StataCorp LP, College Station, TX, USA).

## 3. Results

### 3.1. Study Selection

Figure 1 presents the flow diagram of study selection. We identified 3226 studies by searching relevant databases and 5 studies through manually searching the references of relevant papers, resulting in a total of 3331 studies. After the exclusion of duplicate records and nonrelevant studies, as determined after the screening of titles and abstracts, a total of 61 studies were selected at the full-text review stage. Of these 61 studies, 44 were excluded (12 were case reports or series and letters, 1 was a cadaver study, 1 was a review, 6 excluded the target population, 6 did not include the index test of interest, and 7 did not examine the outcome of interest). Finally, 17 studies that met the inclusion criteria were included [4,7,8,9,10,11,12,13,14,15,16,17,18,19,20,21,22].

### 3.2. Study Characteristics

Table 1 summarizes the characteristics of studies included in the meta-analysis. Among 17 studies, 14 [4,7,9,11,12,13,14,15,16,17,18,19,21,22] were performed in Asia, 2 [8,20] in Europe, and 1 [10] in Australia. A total of 5 studies [4,16,19,20,21] were prospective, 11 [8,9,10,11,12,13,14,15,17,18,22] were retrospective, and 1 [7] was a case–control study. Regarding the study setting, six studies recruited patients only from the ED, three recruited only hospitalized patients, three included patients from the ED or outpatient department or hospitalized patients, and five did not provide clear information. Regarding inclusion criteria, seven studies [4,8,10,13,17,21,22] focused on the ingestion of fish bones and seven [7,9,12,15,16,17,21] focused on esophageal FBs. The sample sizes of studies ranged from 45 to 1338. The mean age of included participants ranged from 5.2 to 57 years. Eight studies [7,9,11,12,13,15,19,21] recruited more men than women, seven [4,8,10,14,17,18,20] recruited more women than men, and two did not provide information regarding the number of men and women included in the study. In terms of the index test, eight studies used conventional radiography, including anterior–posterior and lateral cervical X-ray, posterior–anterior and lateral chest X-ray, and abdominal X-ray. In the remaining nine studies [4,7,8,9,12,14,16,18,22], only lateral neck X-ray was used. Regarding the reference standard, all studies used different types of endoscopy techniques, and in six studies [4,13,14,17,20,21], endoscopy was combined with a clinical follow-up. The type and location of ingested FBs differed among the included studies, and related details are provided in Appendix A. The methodological quality of eligible studies is presented in Appendix A.

### 3.3. Overall Diagnostic Meta-Analysis

A total of 17 studies [4,7,8,9,10,11,12,13,14,15,16,17,18,19,20,21,22] (*n* = 4809) reporting diagnostic parameters for detecting FBs in the upper digestive tract were pooled in the meta-analysis. The results revealed that the pooled sensitivity and specificity of conventional X-ray were 0.58 (95% CI = 0.36–0.77, I^2^ = 98.52; Figure 2) and 0.94 (95% CI = 0.87–0.98, I^2^ = 94.49; Figure 2), respectively, for detecting FBs in the upper digestive tract. A high heterogeneity was observed among studies. In addition, the pooled PLR, NLR, and DOR were 10.1 (95% CI = 4.4–23.3; Appendix A), 0.44 (95% CI = 0.27–0.74; Appendix A), and 23 (95% CI = 7–70; Appendix A), respectively. The area under the SROC curve demonstrated a high accuracy of conventional X-ray in detecting FBs in the upper digestive tract (0.91, 95% CI = 0.88–0.93; Figure 3). The results of Deeks’ funnel plot asymmetry test revealed no significant publication bias (*p* = 0.41; Appendix A).

### 3.4. Subgroup Analysis

We performed a subgroup analysis based on potential factors that may affect the diagnostic accuracy. The results are summarized in Table 2. No significant differences in the sensitivity of conventional X-rays were observed among studies conducted in different continents, those including different types of FBs, those conducted in different settings, and those with different designs. However, the sensitivity of conventional X-ray was significantly higher when used to detect FBs in the esophagus than when used to detect FBs in not only esophageal locations (esophageal FBs = 0.85 and not only esophageal FBs = 0.35; *p* = 0.01). No significant subgroup differences in the specificity of conventional X-rays were noted for the aforementioned factors. Moreover, for the subgroup based on the age, nine studies [4,7,8,10,11,12,14,17,21] focused on adults, one study [19] focused on pediatrics, and seven studies [5,6,9,13,15,16] focused patients of all ages. However, in those studies that focused on patients of all ages, no subgroup data based on adults or pediatrics were reported. Therefore, only results on the adult subgroup were pooled. The results indicated that the pooled sensitivity and specificity of conventional X-ray were 0.48 and 0.88, respectively. In contrast, the result from the only study evaluating pediatrics by Wai Pak et al. [19] showed the sensitivity and specificity of conventional X-rays were 0.16 and 0.99, respectively.

We performed a sensitivity analysis by excluding five studies [8,10,13,17,21] with a sample size of <100 to determine whether the results were influenced by the potential overestimate of the diagnostic performance from the studies with small sample sizes. We observed no significant effect of the sample size on results, with a pooled sensitivity of 0.62 (95% CI = 0.30–0.86) and a pooled specificity of 0.92 (95% CI = 0.84–0.96). The results are summarized in Appendix A.

## 4. Discussion

To our knowledge, this is the first systematic review and meta-analysis to investigate the diagnostic performance of conventional X-rays for detecting ingested FBs in the upper digestive tract. The results of our meta-analysis demonstrated that the conventional X-ray has a sensitivity of 58% and a specificity of 94% for detecting FBs in the upper digestive tract. The area under the SROC curve was 0.91.

The sensitivity of a tool is to measure the proportion of positives that are correctly identified. A high sensitivity test is reliable when its result is negative since it rarely misdiagnoses those who have the disease. In our study, we found that the overall sensitivity of the conventional X-ray for detecting FBs in the upper digestive tract was low. This low sensitivity of conventional X-ray can be attributed to multiple factors. First, the sensitivity may be affected by the FB type, which varies among age groups. In children, coins are the most commonly ingested FB [26]. However, in adults, the most commonly ingested FBs are fish bones (9–45%), other bones (8–40%), and dentures (4–18%) [27]. Second, many fish bones were detected in patients using normal X-ray, but the radiopacity of fish bones is poor in certain fish species [28]. In our study, we explored this factor by subgrouping patients with only fish bones or those with fish bones and others. The sensitivity of conventional X-ray was low in both these subgroups, with no difference noted between the subgroups. In addition, we examined whether the sensitivity of conventional X-ray is affected by the inclusion of patients from different continents, different study settings, and the study design. We did not observe differences between studies including patients from Asia or those including patients from continents other than Asia, between studies including patients from the ED and those not only including patients from the ED, or between prospective and non-prospective studies.

FBs may not be observed when viewed against a bone or a dense soft tissue in the background, such as in the oropharynx and hypopharynx. In other words, the sensitivity of conventional X-rays could be affected by the location of FBs. In our subgroup analysis, we found that conventional X-ray showed higher sensitivity when used to detect esophageal FBs (sensitivity of 85%) than when used to detect no only esophageal FBs. However, this finding should be interpreted with caution, considering that most of these patients underwent conventional X-ray after the ENT consultation, and the flexible endoscopy result was negative [9,10,11,12,19,21]. In addition, these patients may have developed persistent or more severe symptoms, and hospitalization might have been arranged to perform rigid esophagoscopy under general anesthesia. In these patients, FB-related radiographic signs may frequently occur, including the presence of radiopaque density, air accumulation, and soft tissue swelling and loss of cervical lordosis. Luo et al. reported that the sensitivity of conventional X-rays increased with the number of signs combined and interpreted together [7]. On the basis of these findings, we do not suggest using conventional X-rays to exclude patients without FBs in the upper digestive tract.

Ruling in the presence of FBs in the upper digestive tract of patients is also crucial. Multiple overlapping structures of the soft tissue structures and variable patterns of laryngeal cartilage calcification can masquerade as FBs in the upper digestive tract [29]. By contrast, because most FBs, except for some radiolucent materials such as the bones of certain fish species, wood, and plastics, have higher densities than soft tissues and absorb more X-ray photons, they are more radiopaque [30]. Our study results showed that conventional X-ray has a specificity of 94% for detecting FBs in the upper digestive tract. This high specificity may be attributed to the type of FB, which most clinicians can correctly identify. In addition, the results implied that most attending physicians, otolaryngologists, and radiologists rarely fall into the common pitfall, originating from the normal variation of laryngeal cartilage calcification. Our subgroup analysis revealed that the specificity of the conventional X-ray was high, irrespective of whether the included FB was a fish bone, the location of the FB was the esophagus, the patients were from different continents or different study settings, and the studies were prospective or non-prospective. The overall pooled PLR was 10.1, indicating that clinicians could confirm the presence of FBs in the upper digestive tract when conventional X-ray showed positive results.

For clinical application, FBs are commonly lodged in the oropharynx and posterior hypopharynx, which may be detected through laryngoscopy. In most related studies, conventional X-ray was used after the FB was not detected through laryngoscopy as concern regarding the presence of FBs has persisted [7,9,10,11,17,19,20,21]. Therefore, applying this strategy can be reasonable. The main clinical problem in our study was that we observed 42% falsely negative cases after conventional X-ray examinations. Considering that 80–90% of FBs pass through the gastrointestinal tract spontaneously and that potential anesthetic risks and discomfort arise when further esophagoscopy or endoscopy is employed, the observation of the clinical course for a short period was reasonable. Further esophagoscopy or endoscopy should be reserved for patients with persistent or deteriorated clinical symptoms and when suspicion arises of the ingestion of sharp or pointed FBs that can increase the risk of perforation. In addition, studies have reported that computed tomography (CT) has satisfactory sensitivity, ranging from 85.7% to 100%, and specificity, ranging from 66.7% to 100% [10,11,13,17,21]. However, its high cost and radiation exposure limit its application as an initial screening tool. CT can provide clear information regarding the location of FBs and the related complications they cause. Therefore, it can be used as the second step and should be urgently performed when patients have symptoms and signs that suggest perforation or other complications that may require surgery.

This study had several limitations that should be addressed. Firstly, of the 17 studies, 11 [8,9,10,11,12,13,14,15,17,18,22] were retrospective, which may bias the estimate. However, we found that our findings were not affected by the study design through the subgroup analyses. Secondly, most of the included studies were rated as having a high risk of bias because patients were not enrolled consecutively or randomly, and no clear information regarding the cutoff value of the index test or the use of multiple reference standards was provided. Thirdly, we found considerable heterogeneity in pooled sensitivity and specificity, and this could attribute to the different types of FBs, different locations of FBs, and different disease spectrum of patients included in our meta-analysis. Fourthly, the influence of the technical details of the image, the density of FB, and the sizes of FB cannot be determined since most studies did not provide any information. Fifthly,14 [4,7,9,11,12,13,14,15,16,17,18,19,21,22] of the 17 studies were conducted in Asia. Therefore, whether the findings of this study are applicable to patients from other continents should be investigated in future studies. Finally, only one study evaluating pediatric patients by Wai Pak et al. [19] showed the sensitivity and specificity of conventional X-rays were 0.16 and 0.99, respectively. Of note, the sensitivity and specificity of conventional X-ray remained low and high, respectively. Altogether, further well-designed prospective studies are warranted to clarify these limitations.

Overall, despite the aforementioned limitations, our study results support the recommendations of the ESGE guideline [5] for using a conventional X-ray to detect ingested FBs. Another strength of our study is that it is, to the best of our knowledge, the first meta-analysis to include a large sample size (N = 4809) and investigate this crucial clinical issue. In addition, no publication bias was detected from Deeks’ funnel plot asymmetry test.

## 5. Conclusions

The overall sensitivity and specificity of conventional X-rays were low and high, respectively, when used to detect FBs in the upper digestive tract. Thus, we recommend not using a conventional X-ray to exclude patients without FBs in the upper digestive tract. Additional imaging studies or endoscopy examinations should be performed for at-risk patients.

## Figures and Tables

**Figure 1 diagnostics-11-00790-f001:**
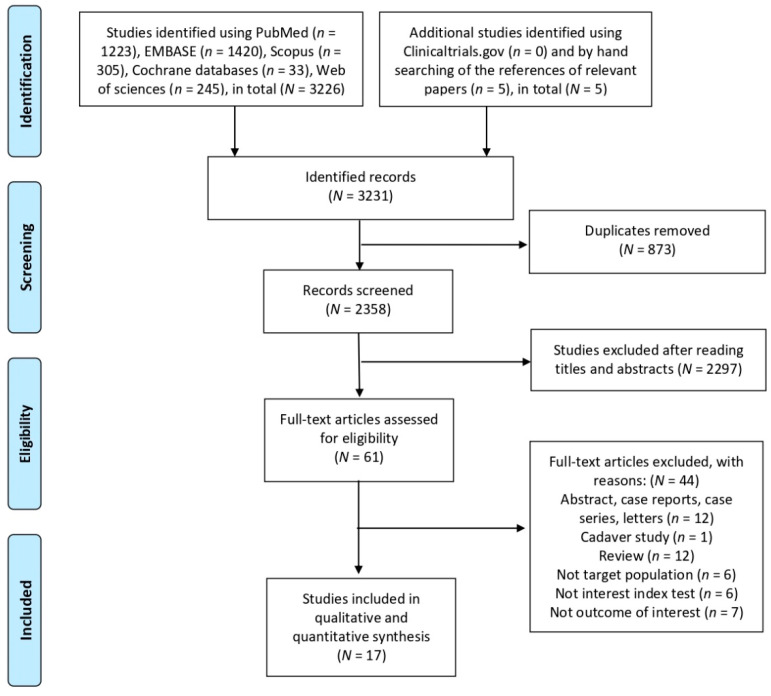
Flowchart of study selection for the current meta-analysis.

**Figure 2 diagnostics-11-00790-f002:**
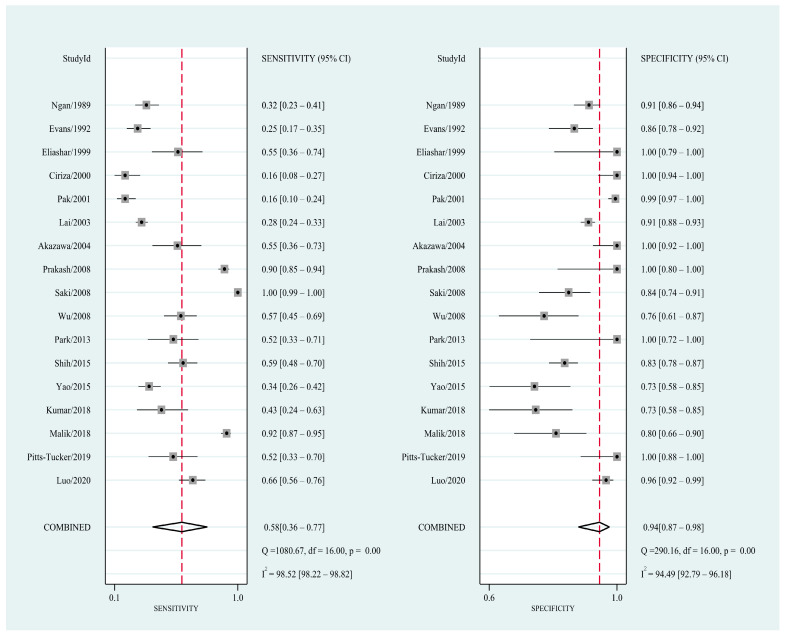
Forest plot of the sensitivity and specificity of conventional X-rays for the detection of foreign bodies in the upper digestive tract.

**Figure 3 diagnostics-11-00790-f003:**
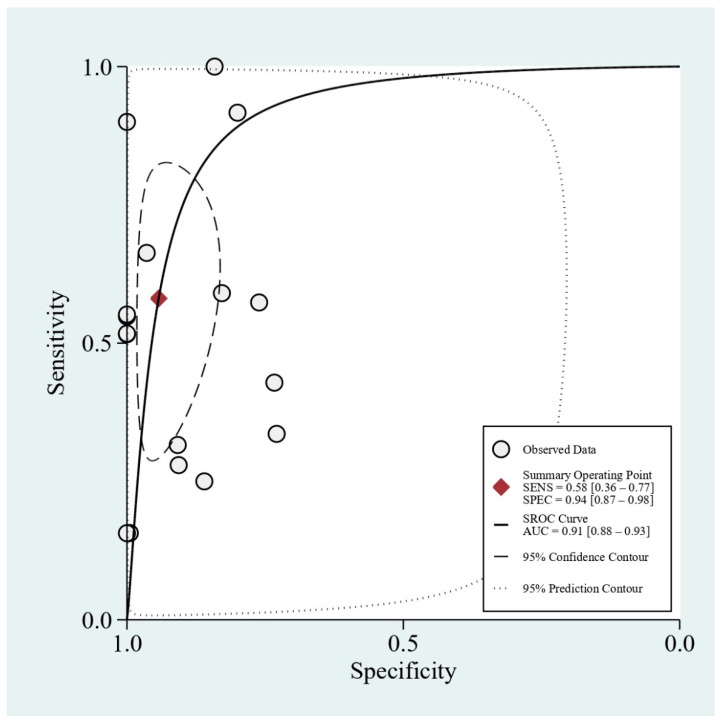
Area under the summary receiver operating characteristic curve of conventional X-rays for detecting FBs in the upper digestive tract.

**Table 1 diagnostics-11-00790-t001:** Characteristics of studies included in the meta-analysis.

Study, Publication Year	Country	Design	Setting	Inclusion Criteria	Sample Sizes	Age, Year (Mean ± SD)	Sex (F/M)	Index Test	Reference Standard
Luo et al. [7], 2020	Taiwan	Case–control study	ED	Suspected esophageal FBs Age ≥ 18 y	235	55.3 ± 16.2	117/118	Lateral neck X-ray	Rigid esophagoscopy
Pitts-Tucker et al. [8], 2019	UK	Retrospective	ED or OPD	Suspected fish bone ingestion	86	46 ± NA	47/39	Lateral neck X-ray	Flexible transnasal endoscopy
Malik et al. [9], 2018	Pakistan	Retrospective	ED or OPD	Suspected esophageal FB ingestion	290	12.4 ± 8.1	121/169	Lateral neck X-ray	Rigid esophagoscopy
Kumar et al. [10], 2018	Australia	Retrospective	ED	Suspected fish bone ingestion	73	50.2 ± NA	51/22	Conventional radiography	Flexible transnasal endoscopy and laryngoscopy
Yao et al. [11], 2015	Taiwan	Retrospective	ED, OPD, or hospitalization	Suspected FB ingestion Age > 17 y	198	57 ± 16	78/120	Conventional radiography	Flexible endoscopy
Shih et al. [12], 2015	Taiwan	Retrospective	ED	Suspected esophageal FB ingestion	345	52.02 ± 12.4	156/189	Lateral neck X-ray	Rigid esophagoscopy and flexible transnasal esophagoscopy
Park et al. [13], 2014	Korea	Retrospective	NA	Suspected fish bone ingestion	66	48.7 ± NA (range: 6–72)	32/34	Neck X-ray: AP, lateral	Endoscopy and clinical follow-up
Wu et al. [14], 2008	Taiwan	Retrospective	NA	Suspected FB ingestion Age ≥ 12 y	114	46.6 ± NA (range: 17–82)	66/48	Lateral neck X-ray	Fibre-optic laryngoscopy, rigid esophagoscopy, and clinical follow-up for 1 month
Saki et al. [15], 2008	Iran	Retrospective	Hospitalization	Suspected esophageal FB Age > 15 y	705	NA; median 47.5 (range: 15–78)	177/528	Conventional radiography	Rigid esophagoscopy
Prakash et al. [16], 2008	Nepal	Prospective	Hospitalization	Suspected esophageal FB	247	NA (range: 9 m–74 y)	NA	Lateral neck X-ray	Rigid esophagoscopy
Akazawa et al. [17], 2004	Japan	Retrospective	NA	Suspected esophageal fish bone ingestion	76	49.8 ± NA	49/27	Conventional radiography	Rigid esophagoscopy or endoscopy; clinical follow-up for 1 week
Lai et al. [18], 2003	Hong Kong	Retrospective	ED	Suspected FB ingestion	1338	43 ± NA (range: 7–98)	719/619	Lateral neck X-ray	Flexible esophagogastroduodenoscopy
Wai Pak et al. [19], 2001	Hong Kong	Prospective	ED	Suspected FB ingestion Age ≤ 12 y	311	5.2 ± 4.1	130 girls/181 boys	Conventional radiography	Tongue depressor, flexible laryngoscopy, Macintosh laryngoscopy, indirect laryngoscopy, and rigid esophagoscopy
Ciriza et al. [20], 2000	Spain	Prospective	ED	Suspected FB ingestion or food bolus impaction	122	54 ± NA (range: 1–91)	72/50	Conventional radiography	Endoscopy, follow-up for 24 h
Eliashar et al. [21], 1999	Israel	Prospective	NA	Suspected esophageal fish bone or chicken bone ingestion	45	55.0 ± NA (range: 31–87)	22/23	Conventional radiography	Rigid esophagoscopy and clinical follow-up
Evans et al. [22], 1992	Hong Kong	Retrospective	NA	Suspected fish bone ingestion	200	NA (range:10 m–91 y)	NA	Lateral neck X-ray	Flexible endoscopy
Ngan et al. [4], 1989	Hong Kong	Prospective	Hospitalization	Suspected fish bone ingestion	358	41.5 ± 16.9 (range: 12 m–86 y)	204/154	Lateral neck X-ray	Flexible endoscopy and clinical follow-up

ED, emergency department; F, female; FB, foreign body; M, male; NA, not available; OPD, outpatient department; SD, standard deviation.

**Table 2 diagnostics-11-00790-t002:** Subgroup analysis.

Subgroup	No. of Studies	Pooled Sensitivity (95% CI)	*p* Value	Pooled Specificity (95% CI)	*p* Value
Continent					
Asia	14	0.63 (0.41–0.85)	0.27	0.94 (0.88–0.99)	0.80
Non-Asia	3	0.34 (−0.12–0.81)		0.97 (0.91–1.00)	
Types of FBs					
Fish bone	7	0.44 (0.11–0.77)	0.30	0.96 (0.90–1.00)	0.74
Not only fish bone	10	0.67 (0.42–0.92)		0.93 (0.86–1.00)	
Location					
Esophageal	7	0.85 (0.72–0.98)	0.01 *	0.95 (0.89–1.00)	0.92
Not only esophageal	10	0.33 (0.15–0.52)		0.93 (0.87–1.00)	
Setting					
Only ED	6	0.35 (0.04–0.67)	0.12	0.95 (0.89–1.00)	0.89
Not only ED	11	0.70 (0.48–0.91)		0.94 (0.87–1.00)	
Design					
Prospective	5	0.41 (0.03–0.80)	0.32	0.99 (0.97–1.00)	0.07
Non-prospective	12	0.65 (0.41–0.88)		0.90 (0.83–0.96)	
Age					
Adult	9	0.48 (0.44–0.52)	NA	0.88 (0.86–0.91)	NA

CI, confidence interval; ED, emergency department; NA, not applied; * statistically significant.3.5. Sensitivity Analysis.

## Data Availability

Available at this manuscript.

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
