# Peer review of "Diagnostic Performance of Conventional X-ray for Detecting Foreign Bodies in the Upper Digestive Tract: A Systematic Review and Diagnostic Meta-Analysis"

_diagnostics, 2021, doi:10.3390/diagnostics11050790_

Round 1
Reviewer 1 Report
Dear Authors, I greatly appreciated your Systematic Review and Meta-Analysis regarding the Diagnostic Performance of Conventional X-ray for Detecting Foreign Bodies in the Upper Digestive Tract.
The paper is well structured and exposed, with a precise explanation of the methods, a complete discussion and good language.
I have just one question regarding your results: have you noticed any subgroup differences in the diagnostic accuracy in relation with age (pediatric/adult)?
Regarding your discussion and comments, it would be interesting to add suggestions regarding the type of study that should be designed to better study this aspect of x-ray diagnosis. You are the first ones who gathered important data about this subject: what would you suggest to giude further research in the field?
Thank you.
Reviewer 2 Report
Review diagnostics-1163394
Diagnostic performance of conventional X-ray for detecting foreign bodies in the upper digestive tract: A systematic review and diagnostic meta-analysis
Introduction
P2l47: please specify: FB usually are no problem at all since they will just pass the intestine after being swallowed. The problem are obstructing FB that do not allow for normal passage.
P2l57: detection of FB is crucial an usually urgent depending on their location. The paper you cited distinguishes urgent and emergent. Maybe you should ahere to this.
P2l64 : this sensitivity is of course due to FB characteristics (radiodense vs lucence)
Methods
P2l85 : there’s a lack of inclusion criteria of either adults or children. They should not be mixed together since completely different populations and different acquistion techniques for radiographic assessment. If applicable adapt you title.
Results
Table1/S2 : basically it is a metaanalysis for the detection of swallowed bony structures, mostly fish bones. Maybe you should think of condensing your results and only focus on bony/ calcified/ radiodense structures. FB others than metal/ bone e.g. glass won’t be depictable on radiographs anyway.
P4fig1 : what does no target population mean ?
P8l161 : same applies here. Since the studies nearly only report fish or other bones it is not correct to generalize the results to all FB, since glass and plastic is not covered sufficiently.
P8l165 : are you allowed to pool the studies when having high heterogeneity ? Was this checked by a statistician?
P10tab2 : In your subgroup analysis : does types of FBs – not only fish bone include fish bone ? it should be bone structures excluded (what is the number of studies ? n=xxx). If you are not able to give comprehensible results for non-bony FB, let it fall and concentrate on bones only.
Overall : since you focus on radiography for FB please give some technical details on image acquisition of the studies included. Are details for image acquisition/ acquisition parameters given ? CR/ DR/ film radiography ?
P10l191 : I did not understand why you excluded studies with sample size <100 ?
Discussion
Needs major restructure and additional work.
P10l196: this is for sure not the first systematic review (e.g. ref 27).
P10l199: again, give subgroup analysis for bony vs non-bony FB or concentrate on bony only if applicable. Either children or adults, please don’t mix them up.
P10l101 : I guess you mean : »a tool used to correctly identify patients with FB ? Strange formulation.
P10l204 : the sensitivity is not only influenced by the type of FB it is dependend on radiolucency. High density FB will be identified easily, low density or transparent one very hard. Size is another important factor. Again for children usually another x-ray quality is used for image acquisition that will impact image quality compared to an adult.
P10l209: Thanks for ref28, made me smile, when I read this paper J
P10l210: again, what do you mean with «not only fish bones»?
P10general: since you publish on a radiology topic you have to discuss technical background. E.g. about differences in acquisition parameters of the studies you included.
P10l218 : I do not agree. A trained radiologist should identify the FB, missing FB due to summation of anatomic structures and the FB is of course possible but I think it’s wrong to attribute the low sensitivity to a bad performance of the radiologist. Maybe you can use these papers to restructure your discussion (https://www.ajronline.org/doi/pdf/10.2214/AJR.13.12185, https://pubs.rsna.org/doi/full/10.1148/rg.233025137). You will find a large amount of radiology papers that I miss a little bit in you references, only the last ones.
P11l263 : this statement does not reflect current recommendations. See e.g. https://www.ajronline.org/doi/pdf/10.2214/AJR.13.12185
p11l279: this is not what the ESGE guidelines say. Similar as the german ones for children (ref 27) they are a bit more complex. ESGE does not recommend radiological evaluation
for patients with nonbony food bolus impaction without complications.

Round 2
Reviewer 2 Report
all querys and questiond answered satisfactory.